# A Scoping Review of the Clinical Evidence for the Health Benefits of Culinary Doses of Herbs and Spices for the Prevention and Treatment of Metabolic Syndrome

**DOI:** 10.3390/nu15234867

**Published:** 2023-11-22

**Authors:** Marion Mackonochie, Ana Rodriguez-Mateos, Simon Mills, Vivien Rolfe

**Affiliations:** 1Pukka Herbs Ltd., 10 York Road, London SE1 7ND, UK; simon.mills@pukkaherbs.com (S.M.); vivien.rolfe@gmail.com (V.R.); 2Department of Nutritional Sciences, School of Life Course and Population Sciences, Faculty of Life Sciences and Medicine, Kings College London, London SE1 9NH, UK; ana.rodriguez-mateos@kcl.ac.uk

**Keywords:** diabetes, herbs and spices, metabolic syndrome, nutrition, phytochemicals, preventative health

## Abstract

Metabolic syndrome (MetS) is a growing global health problem. Evidence suggests that diets rich in phytochemical-containing herbs and spices can contribute to reducing the risk of chronic diseases. This review assesses the scope of evidence supporting the use of herbs and spices in the diet for the prevention or treatment of MetS and its associated health conditions. A search of the PubMed, Scopus and Google Scholar databases was carried out to assess the available clinical evidence for culinary doses of commonly used herbs and spices. Trials that were measuring health factors related to metabolic disorders in healthy individuals, or the health of individuals with MetS or associated diseases, were included. Out of a total of 1738 papers identified, there were 142 relevant studies on black pepper, chilli, cardamom, cinnamon, coriander, cumin, fennel, fenugreek, garlic, ginger, nigella seed, rosemary, sage and turmeric. No relevant research was found for cloves, mint, oregano, parsley or thyme. Cinnamon, fenugreek and ginger were the herbs/spices with the most published trials on them and that showed promise for glycaemic control. Cardamom appears to have potential to reduce inflammatory markers, and cinnamon, ginger and turmeric to reduce blood lipids. Patients with type 2 diabetes were the population most likely to be included in studies, but the preventative benefits of herbs/spices in healthy populations were also investigated, particularly for chilli, ginger and cinnamon. There is evidence for the beneficial effect of culinary doses of many common herbs/spices in the prevention and treatment of MetS and associated disorders.

## 1. Introduction

Metabolic syndrome (MetS) and its associated conditions, such as obesity, diabetes and cardiovascular disease, are a growing global health problem. Between 2000 and 2019, global diabetes rates grew by more than 1.5% annually and prevalence rates for all other metabolic diseases also increased [1]. Poor diet and physical inactivity are risk factors for the development of MetS and lifestyle changes are key for treatment [2]. MetS involves the dysregulation of blood glucose, insulin resistance, raised blood lipids, increased inflammation and high blood pressure [2]. Therefore, measuring these biomarkers in healthy individuals can indicate the risk of MetS developing or can be used to monitor progress in those with MetS.

Research indicates that the inclusion of herbs and spices in the diet, as is often the case in Mediterranean and Asian diets, may contribute to positive long-term health outcomes [3,4]. Herbs and spices are a particularly rich source of phytochemicals and the consumption of diets rich in phytochemicals has been linked with a reduced risk of cardiometabolic disease and obesity [5,6].

Many studies looking at the health benefits of herbs and spices use high-dose extracts; however, these do not reflect the main way that the general public might be able to take advantage of these relatively cheap additions to their diet. Are these more expensive high-dose formulations necessary for everyone to benefit from herbs and spices, or do culinary doses provide benefits too?

Some researchers have begun to investigate this question. Clinical studies using herb and spice mixes to improve physiological responses to food have indicated that the inclusion of herbs/spices in the diet may have preventative or therapeutic benefits [7,8,9,10,11,12,13,14,15]. The spice mixes used in these studies ranged in dose from 6 g to 23.5 g and positively impacted vascular function, blood glucose and insulin and blood lipids following meals. These effects may contribute to a reduced risk of MetS and its associated conditions when herbs and spices are consumed regularly. The reasons for the specific herb/spice mixes chosen and doses used were often not explained. Doses of 6 g of Italian herb mixes [8], 23.5 g of Asian spices [11] or a combination of Mediterranean herbs and Asian spices at doses of 14.5 g [10] and between 0.5 and 6.6 g [13,14] have been found to have benefits. The lack of consistency in herb/spice mix formulations makes it challenging to attribute benefits to a particular herb/spice, a combination of herb/spices or a dose.

Zanzer et al. assessed the effect of the concentrated liquid extracts of individual spices, but standardized them to equal polyphenol contents, enabling the different effects from each spice to be elucidated [15]. The dose of polyphenols provided from each extract corresponded to the amount found in 6 g of cinnamon. Cinnamon and turmeric positively impacted on blood sugar levels, and turmeric reduced appetite; however, ginger and star anise did not have any effect. Therefore, the individual effects from different herbs goes beyond a general benefit from polyphenol intake and needs to be clarified.

The purpose of a scoping review is to identify a body of evidence, explore how it has been reported, identify evidence gaps [16], and perhaps to inform the development of future systematic reviews [17]. The many different types of evidence available for the health benefits of herbs/spices mean that systematic review methods are not yet appropriate. This methodology uses a systematic approach to searching, but produces qualitative results to highlight available research and provide a base for determining what further research is needed.

Therefore, the aim of this review was to assess the clinical evidence available for the metabolic health benefits of culinary doses of a range of common herbs and spices and to investigate whether there was consistency in the doses used and outcomes found. The most promising herbs and spices in the diet for specific health outcomes or populations can be identified, as well as the future clinical research needed to confirm this effect.

## 2. Materials and Methods

A number of previous reviews have identified herbs/spices with beneficial evidence for general health and for MetS and its associated disorders more specifically [2,3,18,19,20,21,22,23,24]. These were assessed to determine a list of herbs/spices that were most likely to have adequate evidence for this scoping review: black pepper (*Piper nigrum* L.), cardamom (*Elletaria cardamomum* (L.) Maton), chilli (*Capsicum frutescens* L.), cinnamon (*Cinnamomum* sp.), cloves (*Syzygium aromaticum* (L.) Merr. and L.M.Perry), cumin (*Cuminum cyminum* L.), fennel (*Foeniculum vulgare* Mill.), fenugreek (*Trigonella foenum-graecum* L.), garlic (*Allium sativum* L.), ginger (*Zingiber officinale* Roscoe), mint (*Mentha* sp.), nigella seed (*Nigella sativa* L.), oregano (*Origanum vulgare* L.), parsley (*Petroselinum crispum* (Mill.) Fuss), rosemary (*Salvia rosmarinus* Spenn.), sage (*Salvia officinalis* L.), thyme (*Thymus vulgaris* L.) and turmeric (*Curcuma longa* L.). A scoping review methodology was then used, referring to the preferred reporting items for systematic reviews and meta-analyses extension for scoping reviews [25].

### 2.1. Search Strategy

The PICO (population, intervention, comparison, outcome) strategy was used to formulate search terms. The research question was as follows: do herbs and spices affect symptoms of MetS in healthy or relevant diseased populations? The population was healthy individuals or those with MetS and related disorders. The intervention was single herbs or spices at culinary doses. The comparison was a control or no treatment. The outcome was a change in symptoms associated with MetS or a change in relevant biomarkers: blood glucose, lipids, insulin or inflammatory markers. PubMed and Scopus were searched in January 2023 with no date restrictions applied. A search of PubMed database was carried out using the search terms: (“black pepper” or “*Piper nigrum*” or “black seed” or “black cumin” or “*Nigella sativa*” or cardamom or “*Elettaria cardamomum*” or chilli or “*Capsicum frutescens*” or cinnamon or “*Cinnamomum zeylanicum*” or cloves or “*Syzygium aromaticum*” or coriander or “*Coriandrum sativum*” or cumin or “*Cuminum cyminum*” or fennel or “*Foeniculum vulgare*” or fenugreek or “*Trigonella foenum-graecum*” or garlic or “*Allium sativum*” or ginger or “*Zingiber officinale*” or mint or “*Mentha*” or oregano or “*Origanum vulgare*” or parsley or “*Petroselinum crispum*” or rosemary or “*Rosmarinus officinalis*” or sage or “*Salvia officinalis*” or thyme or “*Thymus vulgaris*” or turmeric or “*Curcuma longa*”) AND (metabol* or diabetes or obesity or cardiovascular or “blood glucose” or “blood sugar” or “blood lipids” or “blood fats” or “blood insulin”). The results were filtered by Clinical Trial as the article type. Each herb/spice was searched for separately in Scopus, with the following search term: TITLE-ABS-KEY ({herb/spice name}) AND TITLE-ABS-KEY (metabolic OR metabolism OR diabetes OR obesity OR cardiovascular) AND TITLE (clinical OR human). Additional studies were found using Google Scholar and hand searching reference lists from relevant reviews.

### 2.2. Inclusion and Exclusion Criteria

The inclusion criteria were the use of a whole herb/spice or powdered/ground herb/spice in food, drinks or encapsulated and at doses that could reasonably be achieved in the diet without negatively impacting palatability. Concentrated extracts or oils and combinations of herbs or spices were excluded. Studies of water infusions or herbal teas were included when the formulation and quantity was what might be reasonably consumed at home. Any studies that administered herbal formulations or more than one herb/spice at a time were excluded; however, those with multiple individual herbs/spices being investigated in separate arms of the study were included. To ensure a broad range of different studies, clinical trials that were measuring biomarkers related to metabolic disorder in healthy individuals or the health of individuals with metabolic disease or related conditions were included. Studies were included regardless of the age or health status of participants, as the intention was to determine the potential for herbs and spices to both prevent and treat metabolic disease. However, if the participants had a co-morbidity not related to MetS the study was excluded. Studies were included regardless of language. Any retrieved studies not in English were translated using Google Translate. Animal and in vitro studies were excluded. Reviews were excluded.

### 2.3. Study Selection and Data Collection

The articles identified from title and abstract screening were added into a Microsoft Excel spreadsheet, the papers were retrieved, and final inclusion decisions were made by reading the full text. Two reviewers (MM and VR) screened this list to decide on the final included studies. For each article, the following data were entered into the spreadsheet: study type, herb/spice investigated, population, dose of herb/spice, length of study and outcome measures. The evidence for each herb/spice was clustered into type of metabolic health measurement investigated, and whether there were positive findings or not. The Jadad Scale was used to score the methodological quality of the clinical trials. The Jadad scale was originally developed for assessing clinical trials in pain research, but has been widely adapted and is considered to offer the best validity and reliability evidence [26]. It has limitations in trials investigating food, due to the difficulty in blinding with fresh food, but was still considered the most appropriate. One point is scored for each of the following: a mention of randomisation; a description of an appropriate randomisation method; a mention of blinding; a description of an appropriate blinding method; and, all participants in the trial being accounted for in the results. Trials that scored 0–2 were considered low quality and those scoring 3–5 were considered high quality.

## 3. Results

The PubMed search identified 792 results, while the Scopus search produced 925 results. An additional 21 papers were found from Google Scholar and reference list scanning. Title and abstract screening led to 254 papers for full-text screening. This led to a total of 142 relevant papers for data extraction (Figure 1).

Evidence was found for black pepper, chilli, cardamom, cinnamon, cloves, coriander, cumin, fennel, fenugreek, garlic, ginger, nigella seed, rosemary, sage and turmeric. No relevant research was found for mint, oregano, parsley or thyme. Table 1 summarises the included studies. The rationale behind the studies excluded at full-text screening is shown in Appendix A.

### 3.1. Black Pepper, Cardamom and Chilli

There was one single-blind cross-over trial on black pepper in healthy adults. A dose of 1.3 g added to a meal had no impact on appetite or thermogenesis [138].

Ten studies (eight double-blind RCTs, one single-blind RCT and one clinical study) looked at the impact of cardamom on inflammation and a range of metabolic markers in individuals with hypertension [27], diabetes [29,32,59], prediabetes [28], poly-cystic ovarian syndrome (PCOS) [34,35] and non-alcoholic fatty liver disease (NAFLD) [30,33]. All the studies on cardamom used 3 g/day for between 8 and 20 weeks. Five out of six studies that investigated inflammatory markers found positive effects [28,30,31,32,34,35], two studies found benefits on blood glucose and insulin and two studies found no benefit from cardamom on blood lipids, while effects on blood pressure were variable.

Eleven clinical studies [9,36,37,38,39,40,41,42,43,44,45] investigated the effects of chilli on appetite, vascular function and blood glucose, insulin and lipids. Apart from one study looking at the effect of chilli on glucose and insulin in pregnant women with gestational diabetes [42], all the included studies were in healthy individuals.

Four of the intervention studies used doses of 30 g of fresh chilli [36,37,38,39], the other seven intervention studies used doses of 0.6 g [40], 1.25 g [42], 3.09 g [44,45] or 5 g [43], a meal with chilli containing 5.82 mg of capsaicinoids [9] or chilli capsules containing 10 mg of capsaicinoids [41].

Appetite and/or thermogenesis or metabolic rate were measured in eight out of thirteen of the studies and five of these found a beneficial effect. There were as many studies finding positive results as those showing no effect for the key metabolic biomarkers of blood glucose, insulin and lipids (see Figure 2).

One cross-over study found an improvement in diet-induced thermogenesis [36] one single-blind cross-over study found increased metabolic rate [44] and one single-blind cross-over study found decreased appetite from chilli [45]; however, four studies found no effect on metabolic rate or energy intake [9,39,40,41]. A reduction in insulin was found in three studies [9,38,42], while an increase was found in one [43]. There was no change in blood glucose levels from two randomized cross-over studies [37,39], while one double-blind RCT [42] and one clinical study [43] found significant reduction in glucose from chilli consumption. The quality of the clinical studies was generally low (8 low and 3 high according to Jadad scores), mainly due to the difficulty in blinding the consumption of chilli in food.

### 3.2. Cinnamon

There were 41 studies looking at the benefits of culinary doses of cinnamon. Ten in healthy individuals [46,48,49,50,54,55,56,57,81,82], nineteen in those with diabetes [47,53,59,60,61,63,64,66,67,68,69,70,71,74,75,76,78,79,84], six in women with PCOS [58,65,72,73,80,83], one in Asian Indians with MetS [77], one in patients with NAFLD [62], one in individuals with impaired glucose tolerance [51], one in sedentary women with obesity [52], one in women with dyslipidaemia [86] and one in prediabetic individuals [166]. Doses ranged from 1 to 6 g, either as a single dose or daily for between 2 weeks and 6 months. Whether a positive effect was seen or not did not appear to correlate with dosage.

Beneficial effects on glucose were found in 6 randomized cross-over studies, 1 randomized trial and 10 double-blind RCTs from doses of 0.5–6 g/day, 100 mL of cinnamon tea or single doses of 5–6 g of cinnamon [46,48,52,53,54,55,56,57,62,73,74,75,76,77,78,79,82], while 7 double-blind RCTs, 3 single-blind RCTs and 1 randomized cross-over study found no effect from a single dose of 1–6 g or 1–1.5 g/day [47,49,51,60,61,63,65,70,71,80,166].

Beneficial effects on insulin were found in 4 randomized cross-over studies, 10 double-blind RCTs and 1 single-blind RCT from doses of 1–3 g/day or single doses of 3–6 g of cinnamon [46,48,49,57,59,62,68,70,73,75,77,78,79,80,83], while 6 double-blind RCTs, 1 single-blind RCT and 1 randomized cross-over study found no effect from doses of 1–1.5 g/day or a single dose of 6 g [47,49,51,53,60,67,71,166].

### 3.3. Coriander Seed, Cumin and Fennel

One single-blind RCT found that 2 g/day of coriander seeds for 40 days improved average body mass index (BMI) from 27.3 to 26.7 and blood lipids (total cholesterol, low-density lipoprotein (LDL) and high-density lipoprotein (HDL)), as well as systolic blood pressure, in patients with hyperlipidaemia [116].

Two clinical studies looked at the benefit of cumin on anthropometric measures, blood insulin and blood lipid levels in overweight adults [85] and women with dyslipidaemia [86]. The randomised clinical trial by Zare et al. used a dose of 6 g/day for 3 months and this reduced all blood lipid measurements, as well as anthropometric measurements of weight, body mass index (BMI), waist circumference and fat [85], while Pishdad et al. used 3 g/day for 8 weeks in a double-blind RCT and found a benefit on total cholesterol, but not LDL or HDL cholesterol [86].

One single-blind crossover study found that a single dose of 2 g of fennel as a tea decreased appetite in healthy women, but did not impact food consumption [87].

### 3.4. Fenugreek

There were twenty studies looking at fenugreek, mainly for its impact on blood glucose, insulin and lipids, in healthy individuals [87,91,92,95], diabetics [89,90,93,94,96,97,98,99,100,101,103,104,105,106], individuals with coronary artery disease [137] and adults with hyperlipidaemia/hypercholesterolaemia [88,102]. Quantities ranged from 2 g up to 100 g, with the majority of studies using 10–15 g/day. Effects on blood insulin, glucose and lipids were promising, with 11 out of 14 studies showing significant positive effects on blood glucose, 7 out of 8 studies finding significant changes in insulin and 6 out of 8 studies significantly improving blood lipids, regardless of dose used. However, excluding low-quality studies reduced the number of studies indicating a benefit on blood glucose (to 4 out of 5 studies) and blood lipid levels (to 1 out of 2 studies).

### 3.5. Garlic

There were ten clinical studies (four double-blind RCTs and six single-blind or randomised clinical studies) looking at the benefits of garlic. Both clinical trials that looked at the effect of garlic on blood pressure (BP) found benefits when participants consumed 20 g or 100 mg/kg bodyweight of fresh garlic daily [107,111]. Most of the studies investigated the impact of garlic on blood lipids. One clinical study was carried out on overweight smokers [112]. Four clinical studies looked at platelet function [110], cholesterol [108,109], immunity and cancer markers [168] in healthy individuals. Two studies looked at multiple outcomes in patients with MetS [107,115], one study investigated NAFLD [114] and two studies looked at individuals with hyperlipidaemia [111,116]. For the interventional studies, doses ranged from 1.6 g to 40 g, with higher doses of fresh garlic compared with dried garlic powder. All but two of the clinical studies looked at blood lipids and seven out of eight found a benefit. Fresh garlic in doses of 100 mg/kg body weight, 20 g or 40 g/day significantly reduced triglycerides in three studies [107,108,111], while two studies found that 1.6 g/day of dried garlic reduced triglycerides [113,115]. Total cholesterol was significantly reduced by 1.6, 2 or 3 g/day dried garlic and 20 g or 40 g of fresh garlic [108,109,111,113,116]. Only one study of overweight participants at risk of cardiovascular disease found no impact of 2.1 g of garlic on blood lipids [112].

### 3.6. Ginger

Out of a total of 24 studies on ginger, 6 were carried out on healthy individuals to look at blood clotting [119], energy intake and appetite [138], thermoregulatory function or thermogenesis and appetite [118,120], anthropometric measurements [128,133], anthropometric measurements and blood glucose and fats [130] and cardiovascular risk factors [137]. There were 11 studies on patients with type 2 diabetes looking at the impact of ginger on blood sugar, insulin and blood lipids [126,134,135], metabolic health and inflammation [125,132], fasting blood glucose and insulin sensitivity [127,129], blood glucose, insulin and inflammation [121], vascular function [122], anthropometric measurements and blood pressure [84] and anthropometric measurements and inflammation [124]. One study looked at the effects of 1.5 g of ginger on anthropometric measurements and insulin resistance in women with PCOS [83]. One study looked at the effect of 3 g/day on blood lipids in individuals with hyperlipidaemia [136]. Two studies looked at the impact of ginger on liver function, anthropometric measurements, blood sugar and inflammatory markers in individuals with NAFLD [117,123]. One looked at the impact of 1 g of ginger daily in obese children with NAFLD [117], while, in the other study, adults were given 1.5 g/day [123]. A pilot study investigated the impact of 1 g/day of ginger on thyroid symptom score, anthropometric measurements, blood glucose and blood lipids in hypothyroid patients [131].

Ginger was used in doses that ranged from 1 g to 10 g of dried powdered ginger for a single dose or up to 12 weeks daily, apart from one study that used 15 g of fresh ginger or 40 g of cooked ginger [119] and another study that used 20 g of fresh ginger [138]. There did not appear to be a correlation between dose and efficacy. All seven studies that investigated the impact of ginger (doses of 1.5–3 g/day) on insulin found a benefit, ten out of twelve found positive effects on blood glucose (doses of 1.2–3 g/day), six out of seven found a positive effect on blood lipids, such as total cholesterol, LDL and triglycerides (doses of 1–3 g) and four out of five studies looking at inflammatory markers (doses of 1.5–3 g) found a benefit. The studies were mainly of high quality (22 out of 24) according to Jadad scoring, with 18 double-blind RCTs, 3 randomized cross-over studies and 3 placebo-controlled study.

### 3.7. Nigella Seeds

There were 17 studies on nigella seeds, mainly investigating their effect on anthropometric measurements, blood glucose, insulin and lipids. One double-blind RCT was carried out on healthy male volunteers [141]. A large double-blind RCT looked at the impact of 1.5 g/day of nigella seeds in 250 healthy men with MetS [165]. One double-blind RCT was carried out on men with obesity [153]. Two studies were carried out with thyroiditis patients [147,150], four in individuals with MetS [139,140,143,152], three in patients with hyperlipidemia/hypercholesterolaemia [144,146,154], three in patients with type 2 diabetes [142,145,151] and two in patients with NAFLD [148,149]. The studies on nigella seeds used between 500 mg and 3 g/day for durations ranging from 4 weeks to 1 year.

Nigella seeds improved anthropometric measurements such as BMI and weight in three out of seven studies (at doses of 2–3 g/day), improved blood glucose in five out of ten studies (at doses of 500 mg–2 g/day), insulin (at 2 g/day) in three out of four studies, blood lipids in seven out of eleven studies (at 500 mg–2 g/day) and inflammatory markers in one out of four studies (at a dose of 2 g/day). Of the 17 studies, 13 were high-quality according to Jadad.

### 3.8. Rosemary, Sage and Turmeric

The one high-quality, double-blind RCT on rosemary found no impact on liver enzymes, anthropometric measurements, fasting blood glucose, insulin and blood lipids from 4 g/day for 8 weeks in patients with NAFLD [155].

One low-quality, non-randomised cross-over study found that drinking 600 mL of sage tea daily for 4 weeks improved lipid profile but had no effect on blood glucose in healthy female volunteers aged 40–50 years [156].

Five out of the eleven studies on turmeric looked at patients with type 2 diabetes [157,159,161,162,164]. One turmeric study was carried out on healthy volunteers to investigate glycaemic effect [158]. Two other studies were carried out on individuals who were stated to be overweight, obese or prediabetic, with no other health issues [166,167]. Two studies looked at the effect of turmeric on NAFLD [163,169]. The studies used between 1 and 3 g/day for between 4 and 12 weeks, or single doses of 1 g [166] and 6 g [158].

All studies were of a high quality according to Jadad. Three out of five studies investigating anthropometric measurements, such as weight and BMI, found some positive effect from turmeric (at a dose of 2.1 g/day). Four out of seven studies found improvements in blood glucose levels (at doses of 2.1–2 g/day), one out of two studies found improvements in insulin (from a dose of 2 g/day) and five out of six studies found improvements in blood lipids such as triglycerides, total cholesterol and LDL (at doses of 2.1–2.4 g/day). Out of three studies looking at inflammatory markers, two found beneficial effects from 2.1 to 2.4 g/day of turmeric powder in capsules.

### 3.9. Herb/Spice Efficacy

Figure 2 identifies the main health markers measured and whether effects were seen for each of the herbs and spices in all studies and only in high-quality studies. Blood glucose and insulin were the most commonly measured markers, followed by blood lipids, then inflammatory markers. Only including high-quality studies did not make a big difference to the pattern of responses seen for glucose, insulin or inflammatory markers. However, the benefits of fenugreek and garlic on blood lipids were not apparent when only high-quality studies were considered.

### 3.10. Adverse Effects

No adverse effects were reported for any of the studies at the doses used.

### 3.11. Study Quality

Studies were scored for quality using Jadad (details on the scores given are provided in Appendix A). There were 40 low-quality studies (28% of the scored studies), 97 high quality studies (66% of the scored studies), and 4 studies for which there was not enough information to score them. Of the 142 studies, 81 were double-blind RCTs, 30 were cross-over clinical studies, 30 were single-blind or parallel clinical studies and 1 was a cross-sectional observational study.

## 4. Discussion

The aim of this scoping review was to assess the clinical evidence available for culinary doses of herbs and spices, what doses are used and in which health conditions, with a view to identifying areas that need further research. In this review, there were a total of 142 studies looking at the effects of black pepper, chilli, cardamom, cinnamon, cloves, coriander, cumin, fennel, fenugreek, garlic, ginger, nigella seed, rosemary, sage and turmeric on metabolic health. Cinnamon, fenugreek and ginger showed the most promise in controlling blood glucose and insulin. Cinnamon, ginger, nigella seed and turmeric were most promising in terms of having beneficial effects on blood lipid levels. Cardamom, ginger and turmeric showed promise for reducing systemic inflammation due to a decrease in inflammatory markers.

Some herbs/spices were more likely to be researched in a specific population or in an investigation of a specific metabolic biomarker. This either indicates increased efficacy or traditional associations and observations stimulating more research in these areas. Findings from in vitro studies or in vivo animal research may also have prompted researchers to focus on a particular herb and effect. For example, despite not having a traditional association with inflammation, the anti-inflammatory effects of cardamom have been found in an animal model [160].

Cardamom was only investigated in individuals with disease associated with MetS and no studies were carried out in healthy individuals. Most studies investigated the impact of cardamom on inflammatory markers. MetS involves increased inflammation; therefore, changes in inflammatory markers are more likely to be observed in those with MetS than healthy individuals. Preclinical research has identified the potential of cardamom for use in inflammation and hyperlipidaemia. Cardamom reduced swelling and downregulated inflammatory cytokines such as cyclo-oxygenase-2 (COX-2) in an animal model [160]. This confirms the finding in this scoping review that cardamom has anti-inflammatory activity. A terpenoid compound from cardamom, 1,8-cineole, prevented lipid oxidation in vitro and lowered serum lipid levels in zebrafish, while cardamom oil at a dose of 3 g/kg reduced total cholesterol, LDL-cholesterol and triglycerides in Wistar rats [170]. This scoping review did not identify any clinical studies showing a beneficial effect of cardamom on blood lipids, which may reflect a failure of animal studies to relate to effects in humans, or may be due to the difference in dose or formulation.

Almost all the studies on chilli were carried out in healthy individuals. Chilli did not have marked effects on the main insulin, glucose, lipids and inflammatory biomarkers, which may reflect the use of healthy populations for most studies, the doses used or difficulty in blinding chilli consumption in food. It showed more promise for impacting thermogenesis, metabolic rate and appetite. The key active compound in chilli is capsaicin and this binds to transient receptor potential vanilloid receptor 1 (TRPV1) to activate metabolic modulators such as peroxisome proliferator activated receptor (PPAR)α and glucagon-like peptide (GLP)1 [171]. Capsaicin has also been found to decrease ghrelin secretion [171], which would explain its appetite-suppressing effects.

The majority of studies on cinnamon were carried out in patients with diabetes, indicating the strong association of this spice with blood sugar control, e.g., references [172,173,174]. This impact on blood sugar is thought to be due to an insulin-mimetic effect and via the inhibition of digestive enzymes, such as α-amylase, in the gastrointestinal tract [172,174]. Cinnamon activates both PPARα and PPARγ, which would explain its effect on glycaemia [175]. However, there was considerable heterogeneity in the effect of cinnamon on blood sugar and blood insulin across the different studies in this review. Cinnamon was more likely to positively impact blood sugar and insulin in healthy individuals than those with type 2 diabetes. Seven out of nine studies (77%) looking at the impact of cinnamon on blood sugar in healthy individuals found a benefit, while only seven out of fifteen studies (47%) looking at the impact of cinnamon on blood sugar in diabetic patients found a benefit. This suggests that it could be of preventative and therapeutic benefit; however, due to the heterogeneity of the results, systematic reviews isolating the impact of the dose and the health of the participants on outcomes would be interesting. There are a number of meta-analyses of cinnamon in diabetes [172,176,177,178,179,180,181,182]; however, there did not appear to be any assessing blood glucose control in healthy individuals.

The effects of cinnamon on blood sugar, insulin and lipids appeared to be quite mixed from both this scoping review and published meta-analyses. Well-researched areas, such as is found with cinnamon, often highlight heterogenous results. This indicates that further research is needed to separate out the factors that impact efficacy, such as dose, presence in a food matrix, population or duration of study. Yu et al. found that dosage did impact efficacy, with a dose of less than 1.2 g significantly reducing fasting blood glucose, when pooled results found no benefit [175]. There have been at least eight meta-analyses of clinical trials of cinnamon looking at LDL-cholesterol. Six found decreases in LDL-cholesterol [175,176,180,183,184,185], but two found no impact on LDL-cholesterol [186,187]. Yu et al. found that the effect of cinnamon on LDL-C was influenced by dose [175], which may explain some of the heterogeneity.

Fennel has been used traditionally for its digestive properties [188], but it has also been found to have anti-inflammatory and antihyperlipidaemic activity [20]. Bae et al. found that fennel did not impact food consumption; however, it did reduce appetite, indicating that further research might be beneficial [87].

The impact of fenugreek was investigated in patients with diabetes in 14 studies and in healthy individuals in 6 studies. The effects on blood glucose and insulin were mainly positive whether in a healthy or diseased population; however, high-quality studies were less likely to find benefits. Larger doses of fenugreek tended to be used, as the beneficial effect was usually considered to be from the soluble fibre in the seeds [189], although some studies suggest an effect of other compounds such as flavonoids, saponins and the alkaloid trigonelline [190,191]. As a relatively mild-flavoured spice, larger amounts are palatable in the diet. The studies using larger quantities incorporated the seed powder into a food matrix, such as being baked into bread; therefore, these were considered culinary doses and were included.

Nine meta-analyses on fenugreek in MetS or associated conditions have been published in the last ten years [189,190,191,192,193,194,195,196,197]. All of those that assessed effects on blood glucose found a benefit, but concerns were raised about the quality of the clinical trials. A review of fenugreek in blood pressure found a dose-dependent effect, with doses greater than 15 g/day for longer than 12 weeks being effective [192]. Neelakantan et al. also found that dose impacted the effect of fenugreek on glycaemia, with doses higher than 5 g being effective [189].

Studies on garlic looked almost exclusively at blood lipid levels, whether in healthy populations or those with metabolic disorders. Garlic has a strong association with cardiovascular health and cholesterol levels [198,199,200]; however, much of the published research has looked at the effect of standardised garlic extracts, rather than its consumption in food. Systematic reviews and umbrella reviews have identified strong potential for garlic in hyperlipidaemia, hypertension and inflammation [198,199,200].

This review has confirmed the benefits of garlic for hyperlipidaemia and indicates that concentrated extracts may not be necessary in order to benefit from some of the positive health effects from garlic, as both lower and higher doses showed efficacy. There was no relationship between dose and size of effect; for example, the reductions in total cholesterol from 2, 20 and 40 g of garlic compared with control were 82, 19 and 51 mg/dL, respectively [108,111,116]. Both low- and high-quality studies, according to Jadad score, showed effectiveness. Garlic and its phytochemicals have been found to have anti-hyperlipidaemic activity via 3-hydroxy-3-methylglutaryl CoA (HMG CoA) inhibition and reduction in cholesterol synthesis; hypotensive activity via angiotensin-converting enzyme (ACE) inhibition, the downregulation of angiotensin II and the stimulation of nitric oxide; and anti-inflammatory/anti-atherosclerotic effects via COX inhibition, the decreased synthesis of thromboxane B2, the decreased production of leukotriene C4 and a reduction in LDL oxidation [201]. There are feasible mechanisms of action for the findings and evidence for the benefits of garlic is building.

Most of the ginger studies in healthy populations looked at thermoregulatory function and appetite rather than metabolic biomarkers of glucose, insulin or lipids. The obvious sensorial heating effects on the body from ginger can explain this choice of research. The activation of TRPV1 by pungent principles in spices such as ginger leads to a sensation of heat in the mouth upon consumption and has been suggested to have a thermogenic effect [202]. Studies in diabetic patients appeared largely positive for blood glucose, insulin and lipid levels. Whether this is via the activation of TRPV1 receptors, via anti-inflammatory effects through the inhibition of COX and lipoxygenase or via another mechanism remains to be investigated.

Both nigella seeds and turmeric were investigated in a range of health conditions, with no predominance of either healthy individuals or those with type 2 diabetes. This reflects the broad range of uses of these herbs/spices in traditional medicine.

The studies on nigella seeds were heterogenous in terms of the populations investigated and effects seen. The health effects are broad and many traditional medicine systems consider it to be a panacea [203,204,205], which has led to a lack of focus for research into its benefits. However, the phytochemical thymoquinone, found in the essential oil of the seed, has been identified as being responsible for some of the health benefits [203]. This may indicate there is likely to be more benefit from the use of nigella seed oil than the seeds. A meta-analysis by Daryabeygi-Khotbehsara et al. found that there was a reduction in triglycerides by nigella seed oil, but not the seeds [206]. Sahebkar et al. found that nigella seed powder was more effective than the oil for reducing blood pressure [207], while Askari et al. found that the oil was more effective than the powder for blood glucose control [208]. It is likely that thymoquinone and other terpenoids are responsible for supporting healthy blood lipid and glucose levels, but other phytochemicals not found in the oil are responsible for the hypotensive effect. This indicates the importance of assessing the effect of culinary uses separately from that of concentrated food supplements or extracts. Future research focusing on whether dose and formulation impact on the effects and efficacy would be of interest.

There are at least 10 meta-analyses on the use of nigella seeds for MetS and associated conditions [206,207,208,209,210,211,212,213,214,215]. These all find benefits of nigella seeds on anthropometric measurements such as body weight, blood lipids, glucose control and inflammation, apart from a review of studies on patients with NAFLD, which found mixed results on blood lipids, inflammatory markers and glucose control [214].

Most studies on turmeric found a beneficial effect, with the most promising areas being inflammation and blood lipids. A meta-review on the health benefits of turmeric by Rolfe et al. identified osteoarthritis and MetS to be the most promising areas of research [216]. Turmeric is well researched for its use in inflammatory conditions such as osteoarthritis; however, most research focuses on high-dose curcumin extracts due to the poor bioavailability of curcumin [217]. Therefore, it is interesting that the relatively low doses of 2 g were found to have some effect in this review. Sahebkar found in a meta-analysis that overall turmeric reduced CRP, but that bioavailability-improved preparations of curcuminoids were superior [218], so it may be that higher doses are preferable but not essential.

Nearly all of the spices investigated, but none of the herbs, had some evidence to support their use in culinary doses for the prevention or treatment of MetS and associated disorders. Four mixed herb/spice intervention studies found metabolic benefits from Italian herb seasoning mixes or mixes containing Mediterranean herbs rosemary, basil, thyme, oregano and parsley [8,10,13,14]; however, this scoping review has found a lack of studies to confirm the effects of individual herbs mint, parsley, thyme, rosemary, sage and oregano. These plants are particularly rich sources of volatile oils; therefore, research investigating the antimicrobial properties of the essential oil is abundant, e.g., references [219,220]. In vitro antimicrobial research is relatively easy to carry out, which may explain why other properties of these herbs have not been investigated to date. In addition to volatile oils, as is the case with the spices cinnamon, turmeric and ginger, these herbs are also good sources of polyphenols [24]. Further research into the general health benefits of adding these herbs to the diet would be beneficial.

Considering the widespread use of black pepper in food, it was surprising that there was only one study investigating the impact of this popular condiment for health. It may be that flavour prevents the use of large enough quantities in food for this spice to be of benefit. One study of black pepper was excluded as it investigated the use of a water extract made from 20 g of black pepper, which was not representative of culinary use [15]. It was not clear from the methodology what final dose was consumed. However, if it was a comparable dose to that used by Gregersen et al. [138], then the two studies found opposite effects on appetite, with Zanzer et al. finding an effect of black pepper on appetite, but Gregersen et al. finding none. Black pepper contains piperine, which is recognised as a phytochemical that enhances the absorption of other food components [221], as well as some prescription drugs [222]. The benefits of black pepper could be largely due to its ability to improve the bioavailability of polyphenols and other phytochemicals in herb/spice mixtures.

Changes in insulin, blood sugar and blood lipids were the most common biomarkers to be investigated. Effective glycaemic control, whether via eating foods with a low glycaemic index or adding in herbs/spices and other phytochemical-containing plant foods that help to reduce the glycaemic index of foods, is crucial for the management and prevention of diabetes, as well as the reduction in cardiometabolic risk factors in diabetics [223]. In terms of coronary heart disease risk, only LDL cholesterol has been proven in formal clinical trials to be a biomarker that can be considered a causative factor. Other factors, such as HDL cholesterol, triacylglycerol, vascular function and oxidative damage, require further evidence before their measurement can be considered predictive [224]. This scoping review found promise for cardamom, cinnamon, fenugreek, garlic, ginger, nigella seeds and turmeric, as there were positive findings from at least five different studies for one or more of these biomarkers. However, whether this translates to clinically significant effects for patients with MetS remains to be tested.

### Limitations and Future Directions

Heterogeneity in the methodology is a major limiting factor in both interpreting the results of this scoping review and the many systematic reviews that have been carried out in this area. As herbs/spices are complex in terms of phytochemistry, as well as effect, and there are not commonly accepted doses, comparisons across different studies are challenging.

The doses used in different studies varied more greatly for herbs/spices which can be used both fresh and dried, such as ginger, garlic and chilli. The difficulty in comparing fresh with dried herbs/spices could be overcome in the future by ensuring that a phytochemical analysis of fresh versus dried samples is carried out. Methodologies that account for differences such as these will add value and enable the clearer comparison of one study with another. Most dried herbs/spices were used in doses of between 1 and 6 g, which is representative of the amounts usually used in cooking (however, the doses were usually chosen based on what amounts had been found to be beneficial in previous studies). A notable exception was fenugreek, which was used in higher doses for the additional fibre benefits. It is possible that investigating higher doses of other herbs/spices may also indicate greater benefit, but this would have to be weighed against palatability.

The combination of multiple herbs and spices is likely to have greater beneficial effect than any individual herb or spice, due to the greater quantity and variety of phytochemicals. A limitation of this scoping review is that focusing on individual herbs prevents comparisons being made with the efficacy of herb/spice mixes. Future research could compare the impact of adding a single herb/spice with that from the synergy of a mix of herbs and spices in the diet, providing a rich variety of phytochemicals.

The duration of the clinical trials is also likely to be a major limiting factor in determining whether consuming herbs and spices in the diet is likely to have a noticeable effect on MetS and cardiovascular health. Although the duration of the studies included in this review did not appear to impact on whether the effect was positive or not, any dietary intervention for preventative health needs to be assessed over a longer period, which adds considerable cost and complexity to trials. Many of these diseases develop over a long period of time, so the next step should be to measure changes in biomarkers associated with them over longer periods. Identifying metabolite markers that indicate an increased consumption of specific herbs may be of use, as has been suggested for measuring the intake of polyphenol-rich foods [225].

The health benefits of herbs/spices are likely due to their phytochemical content and complex interactions between these molecules, other dietary components, the microbiome and the gut wall. Phytochemicals, and polyphenols specifically, have been found to impact carbohydrate absorption and metabolism, gut bacteria populations and the uptake of glucose into muscle and adipose tissue [226]. None of the studies incorporated an analysis of the phytochemical content of the herbs/spices used, which would have been a useful addition to tease out the mechanisms or to ascertain which bioactive phytochemicals are driving the effects. The impact of any dietary intervention is dependent on multiple other factors, such as the rest of the diet, participants’ stress levels and physical activity; therefore, teasing out the true impact of individual herbs/spices remains a challenge.

## 5. Conclusions

Overall, this scoping review has highlighted that there is evidence for the beneficial effect of culinary doses of cardamom, cinnamon, chilli, fenugreek, garlic, ginger, nigella seeds and turmeric in the prevention and treatment of MetS and its associated disorders. Cardamom, ginger and turmeric appear to have the most potential for inflammation linked to MetS, garlic, ginger and turmeric for blood lipids and cinnamon, ginger and fenugreek for blood glucose control. Future research needs to address which factors are most important to unlocking these benefits: the food matrix; the combinations of different herbs/spices; the duration of consumption; and how herb/spice intake interacts with other important dietary and lifestyle changes.

## Figures and Tables

**Figure 1 nutrients-15-04867-f001:**
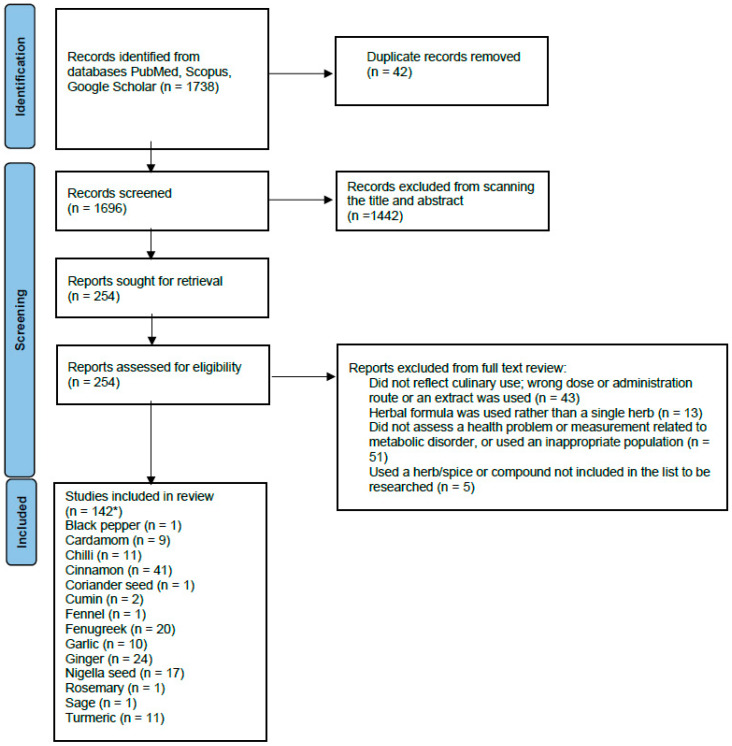
PRISMA flow diagram of the screening process. * The number of studies for all the individual herbs has a total sum of more than 145, as some studies included more than one herb in each arm of the study.

**Figure 2 nutrients-15-04867-f002:**
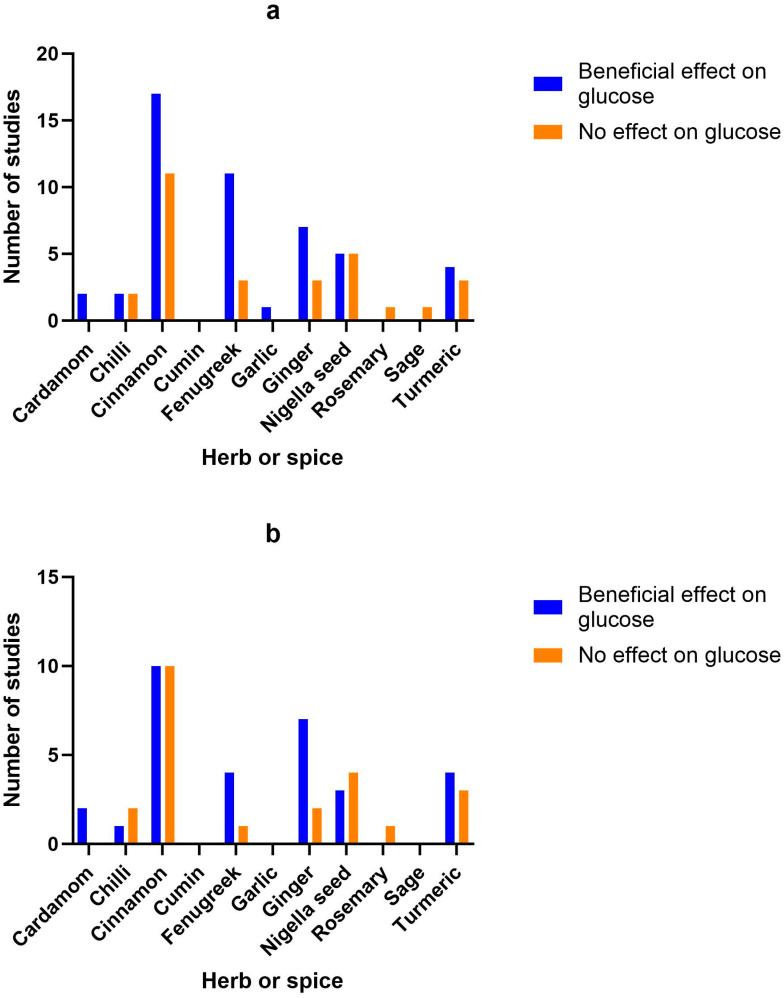
Number of studies and high-quality studies demonstrating the effect of herbs/spices in specific cardiometabolic biomarkers. Figure 2 identifies the main health markers measured and whether effects were seen for each of the herbs and spices in all studies and only in high-quality studies: (**a**) Number of studies showing an effect or lack of effect for each of the herbs and spices on blood glucose; (**b**) Number of high-quality studies showing an effect or lack of effect for each of the herbs and spices on blood glucose; (**c**) Number of studies showing an effect or lack of effect for each of the herbs and spices on insulin; (**d**) Number of high-quality studies showing an effect or lack of effect for each of the herbs and spices on insulin; (**e**) Number of studies showing an effect or lack of effect for each of the herbs and spices on blood lipids; (**f**) Number of high-quality studies showing an effect or lack of effect for each of the herbs and spices on blood lipids; (**g**) Number of studies showing an effect or lack of effect for each of the herbs and spices on inflammatory markers; (**h**) Number of high-quality studies showing an effect or lack of effect for each of the herbs and spices on inflammatory markers.

**Table 1 nutrients-15-04867-t001:** Summary of results from included studies.

Herb or Spice	Reference	Study Type	Population	Dose and Formulation	Length of Intervention	Findings	Study Quality
Cardamom	[27]	Single-blind clinical trial	20 individuals with hypertension	3 g/day in capsules	12 weeks	BP decreased and fibrinolytic activity increased. Fibrinogen and lipid levels did not change.	Low
	[28]	Double-blind RCT	80 prediabetic subjects	3 g/day in capsules	8 weeks	Inflammatory markers CRP, CRP:IL-6 ratio and oxidative stress marker MDA were all decreased.	High
	[29]	Double-blind RCT	83 overweight or obese diabetic pts	3 g/day in capsules	10 weeks	There was an improvement in HbA1c, insulin control and TG and an increase in Sirt1. Cholesterol levels did not change.	High
	[30]	Double-blind RCT	87 overweight or obese pts with NAFLD	3 g/day in capsules	12 weeks	Cardamom improved fatty liver grade, blood glucose, lipids and irisin, but BMI, total cholesterol and FBG were not changed.	High
	[31]	Double-blind RCT	83 pts with type 2 diabetes	3 g/day in capsules	10 weeks	Cardamom led to a decrease in CRP and systolic BP, and an increase in serum NO. Serum ADMA and diastolic BP did not change.	High
	[32]	Double-blind RCT	83 overweight pts with type 2 diabetes	3 g/day in capsules	10 weeks	VCAM, ICAM, E-selectin and IL-6 were decreased in the cardamom group. MMP-9 and CD163 levels were unchanged.	High
	[33]	Double-blind RCT	87 pts with NAFLD	3 g/day in capsules	12 weeks	Cardamom increased Sirt1 and decreased inflammatory markers hs-CRP, IL-6, TNFα and liver marker ALT, as well as improving the degree of fatty liver. Weight, BMI and AST did not change.	High
	[34]	Double-blind RCT	194 obese women with PCOS	3 g/day in capsules	16 weeks	Anthropometric indices decreased. Glycemic indices and androgen hormones improved.	High
	[35]	Double-blind RCT	194 obese women with PCOS	3 g/day in capsules	16 weeks	LH, androstenedione and dehydroepiandrosterone were decreased. FSH increased. Inflammatory markers TNFα, IL-6 and CRP were decreased.	High
Chilli	[36]	Cross-over clinical trial	7 healthy volunteers	30 g fresh chilli	Single dose	A combination of chilli and medium-chain TG increased diet-induced thermogenesis.	Low
	[37]	Randomised cross-over study	36 healthy participants	30 g/day chilli in food	8 weeks	When participants with a BMI > 26 consumed a chilli-containing meal after 4 weeks of daily chilli, there was reduced C-peptide and insulin and higher hepatic clearance of insulin. But blood glucose and energy expended were not changed.	High
	[38]	Randomised cross-over study	27 healthy adults	30 g/day chilli in food	8 weeks	Consumption of chilli increased the resistance of lipoproteins to oxidation, but had no effect on serum lipids, lipoproteins and total antioxidant score.	Low
	[39]	Randomised cross-over study	36 healthy participants	30 g/day chilli in food	8 weeks	There were no effects on metabolic or vascular parameters (glucose, lipids, BP, insulin). However, in men, chilli decreased resting heart rate and increased myocardial perfusion time.	High
	[9]	Randomised cross-over study	34 healthy overweight volunteers	Meal containing chilli with 5.82 mg total capsaicinoids	Single dose	Chilli decreased post-prandial insulin when added to a meal. Metabolic rate, core temperature, CRP and microvascular reactivity were unchanged.	Low
	[40]	Cross-over clinical trial	40 healthy adults	0.6 g in food	Single dose	Eating a meal with chilli in increased the desire to eat sweet food, but had no impact on energy intake	Low
	[41]	Randomised cross-over study	12 healthy adults	10 mg capsaicinoids/day in capsules	5 weeks	There was no change in anthropometric and metabolic measurements from chilli consumption. Chilli increased the firmicutes/bacteroidetes ratio and faecalibacterium abundance that coincided with the increase in plasma levels of GLP-1 and GIP and the decrease in plasma ghrelin level. Benefits were linked to gut enterotypes.	Low
	[42]	Double-blind RCT	42 pregnant women with gestational diabetes	1.25 g/day added to food	4 weeks	Postprandial glucose, insulin and insulin resistance were reduced by chilli. Serum cholesterol and triglyceride were also reduced by chilli. Serum calcitonin gene-related peptide was increased by chilli. When the babies were born, chilli had reduced the incidence of large-for-gestational age newborns.	High
	[43]	Cross-over clinical trial	12 healthy adults	5 g in capsules	Single dose	When 5 g of chilli was administered in a capsule after a glucose challenge, plasma glucose levels were lower after 30 and 45 min than those in the placebo group. Insulin levels in the chilli group were higher than in the placebo group at 1 and 2 h after glucose challenge.	Low
	[44]	Single-blind, randomised, cross-over trial	14 healthy volunteers	3.09 g/day in food	36 h	Chilli increased fat oxidation and prevented reductions in sleeping metabolic rate, diet-induced thermogenesis or resting energy expenditure that were caused by restricting food intake. This indicates a potential beneficial effect in dieting individuals. There was no effect of chilli on BP.	Low
	[45]	Single-blinded, randomised, cross-over design	15 healthy adults	3.09 g/day in food	36 h	Chilli decreased the desire to eat and increased satiety and fullness, particularly when participants under-ate.	Low
Cinnamon	[46]	Single-blind randomised cross-over study	8 sedentary, healthy males	3 g/day in capsules	2 weeks	Cinnamon reduced glucose response to a glucose challenge and improved insulin sensitivity, but effects were not long-lasting once cinnamon consumption ceased.	High
	[47]	Double-blind RCT	25 post-menopausal women with type 2 diabetes	1.5 g/day in capsules	7 weeks	Cinnamon did not improve fasting plasma glucose or insulin concentrations, whole-body oral glucose tolerance or blood lipid profiles.	Low
	[48]	Randomised cross-over study	7 lean, healthy adults	5 g in capsules	Single dose	Cinnamon reduced plasma glucose responses to glucose tolerance tests and improved insulin sensitivity.	Low
	[49]	Randomised cross-over study	15 healthy adults	1 or 3 g/day in food	Single dose	Ingestion of 3 g cinnamon reduced postprandial serum insulin and increased GLP-1 concentrations without affecting blood glucose, GIP, the ghrelin concentration, satiety or GER in healthy subjects. Amounts of 1 g did not have an effect.	High
	[50]	Single-blind randomised cross-over study	9 healthy young adults	3 g in capsules	Single dose	Amounts of 3 g cinnamon did not alter the postprandial response to a high-fat test meal. No change in gastric emptying, glucose response, arterial function, oxidative stress or appetite.	High
	[51]	Double-blind cross-over RCT	10 individuals with impaired glucose tolerance	6 g in capsules	Single dose	No differences in glucose or insulin responses compared with placebo.	High
	[52]	Double-blind cross-over RCT	10 young, sedentary obese women	5 g in capsules	Single dose	Peak blood glucose was lower in the cinnamon group, but blood insulin and insulin sensitivity/resistance were not affected.	High
	[53]	Double-blind RCT	26 pts with type 2 diabetes	1 g/day in capsules	12 weeks	Cinnamon reduced FBG by 6 weeks and this was maintained for the whole 12 weeks of the study. The decrease in HbA1c was not significant. Serum glutathione and superoxide dismutase were increased by cinnamon at 12 weeks, while MDA was reduced, indicating an overall antioxidant effect.	Low
	[54]	Randomised cross-over study	30 healthy obese or normal weight individuals	6 g powder in food	Single dose	Cinnamon reduced blood glucose in obese and healthy weight individuals.	Low
	[55]	Randomised clinical trial	30 healthy adults	100 mL cinnamon tea	Single dose	Cinnamon decreased postprandial maximal glucose level.	Low
	[56]	Crossover clinical study	14 healthy individuals	6 g powder in food	Single dose	Cinnamon reduced the postprandial glucose response and the gastric emptying rate.	Low
	[57]	Randomised crossover study	10 healthy individuals	6 g in food	Single dose	Cassia cinnamon, but not Ceylon cinnamon, reduced postprandial insulin and glucose responses.	Low
	[58]	Double-blind RCT	45 women with PCOS	1.5 g/day in capsules	24 weeks	Menstrual cyclicity improved for women taking cinnamon with no effect on insulin resistance or serum androgens.	High
	[59]	Single-blind RCT	109 adults with diabetes	1 g/day in capsules	12 weeks	Cinnamon lowered HbA1c.	High
	[60]	Single-blind RCT	60 pts with type 2 diabetes	1.5 g/day	12 weeks	Cinnamon had no impact on fasting plasma glucose, HbA1c or serum lipids.	
	[61]	Double-blind RCT	43 individuals with diabetes	1 g/day in capsules	12 weeks	Cinnamon produced no significant change in fasting glucose, lipid, A1C or insulin levels.	High
	[62]	Double-blind RCT	50 pts with NAFLD	1.5 g/day in capsules	12 weeks	There were decreases in HOMA index, fasting blood glucose, total cholesterol, triglyceride, ALT, AST, GGT and high-sensitivity CRP with cinnamon.	High
	[63]	Double-blind RCT	39 adults with diabetes	3 g/day in capsules	8 weeks	Cinnamon had no effect on glycaemic and inflammatory markers.	High
	[64]	Double-blind RCT	44 adults with diabetes	3 g/day in capsules	8 weeks	Cinnamon had no effect on soluble vascular adhesion molecules.	High
	[65]	Single-blind RCT	40 women with PCOS	1.5 g of Ceylon cinnamon in capsules	8 weeks	There was an improvement in cyclicity with cinnamon, which was equivalent to metformin. No change in fasting blood glucose or serum progesterone or androgen levels.	High
	[66]	Double-blind RCT	39 adults with diabetes	3 g/day in capsules	8 weeks	Cinnamon had no effect on inflammatory markers.	High
	[67]	Double-blind RCT	57 adolescents with diabetes	1 g/day in capsules	12 weeks	Cinnamon had no effect on A1c or insulin sensitivity.	High
	[68]	Double-blind RCT	58 pts with type 2 diabetes	2 g/day in capsules	12 weeks	Intake of 2 g of cinnamon reduced the HbA1c, SBP and DBP among poorly controlled type 2 diabetes pts	High
	[69]	Double-blind RCT	59 adults with type 2 diabetes	1.2 g/day in capsules	12 weeks	There was no significant change in SBP from baseline when cinnamon was compared with placebo.	High
	[70]	Double-blind RCT	136 individuals with type 2 diabetes	1.5 g/day in capsules	90 days	HbA1c was reduced by cinnamon, but there was no effect on FBG.	High
	[71]	Double-blind RCT	61 pts with type 2 diabetes	2 g/day in capsules	8 weeks	Cinnamon did not improve FBG, HbA1c, blood lipids.	High
	[72]	Double-blind RCT	84 overweight individuals with PCOS	1.5 g/day in capsules	8 weeks	Cinnamon increased serum antioxidant capacity and improved total cholesterol, LDL and HDL.	High
	[73]	Double-blind RCT	84 overweight individuals with PCOS	1.5 g in capsules	8 weeks	Cinnamon decreased serum FBG, insulin, homeostatic model assessment for insulin resistance, total cholesterol and LDL-cholesterol and weight and increased HDL-cholesterol compared with placebo.	High
	[74]	Triple-blind RCT	105 pts with type 2 diabetes	1 g/day in capsules	12 weeks	Cinnamon improved glucose control and reduced BMI.	High
	[75]	Double-blind RCT	115 pts with type 2 diabetes	0.5 g/day in capsules	12 weeks	Cinnamon reduced FBG, HbA1c and hepatic enzymes. Probiotics were also effective.	High
	[76]	Double-blind RCT	60 people with type 2 diabetes	1, 3, or 6 g/day in capsules	40 days	Intake of 1, 3 or 6 g of cinnamon per day reduces serum glucose, triglyceride, LDL-cholesterol and total cholesterol in people with type 2 diabetes.	High
	[77]	Double-blind RCT	116 Asian Indians with MetS	3 g/day	16 weeks	FBG, HbA1c, waist circumference and BMI were reduced by cinnamon. Waist–hip ratio, BP, serum total cholesterol, LDL-cholesterol, serum triglycerides and HDL-cholesterol were also improved.	High
	[78]	Triple-blind RCT	160 people with type 2 diabetes	3 g/day	12 weeks	Cinnamon reduced HbA1c and blood glucose.	High
	[79]	Triple-blind RCT	140 pts with diabetes	1 g/day	12 weeks	Cinnamon supplementation led to improvement in all anthropometric (BMI, body fat and visceral fat), glycemic (FBG, 2hpp, HbA1C, fasting insulin and insulin resistance) and lipids (cholesterol, LDL-c and HDL-c) outcomes (except for triglycerides).	High
	[80]	Double-blind RCT	59 women with PCOS	1.5 g/day	12 weeks	Fasting insulin, HOMA-IR, LDL and HDL were reduced in the cinnamon group. Changes in blood sugar, serum androgen levels and anthropometric measures were not significant.	High
	[81]	Open, randomised, cross-over clinical trial	21 healthy volunteers	2 g in 200 mL hot water	Single dose	No difference in energy expenditure, dietary-induced thermogenesis, hunger fullness and desire to eat. However, cinnamon tea decreased satiety and increased food intake in the subsequent meal.	Low
	[82]	Randomised cross-over study	18 healthy adults	4 g	Single dose	Cinnamon decreased blood glucose and satiety 15 min after test meal, but did not decrease blood sugar overall.	Low
Cinnamon and ginger	[83]	Double-blind RCT	83 women with PCOS	1.5 g of cinnamon or ginger	8 weeks	Cinnamon and ginger both decreased weight and BMI. Insulin resistance decreased, but only in the cinnamon group. FSH and LH decreased in the ginger group, while testosterone was reduced in the cinnamon group.	High
Cinnamon, cardamom, saffron, ginger	[84]	Single-blind RCT	208 pts with type 2 diabetes	3 g	8 weeks	No difference in BP, serum soluble (s)ICAM-1 concentrations and anthropometric measures.	High
Cumin	[85]	Randomised clinical trial	88 overweight/obese women	6 g/day	12 weeks	Cumin powder reduced serum levels of fasting cholesterol, triglyceride and LDL and increased HDL. Weight, BMI, waist circumference, fat mass and its percentage significantly reduced.	High
Cumin and cinnamon	[86]	Double-blind RCT	99 women with dyslipidemia	3 g/day	8 weeks	Cumin and cinnamon both significantly reduced total cholesterol compared with placebo. Differences in triglycerides, HDL and LDL were not significant.	High
Fennel and fenugreek	[87]	Single-blinded cross-over trial	9 healthy women	2 g fennel infused in 250 mL of water and strained. 24 g of fenugreek, infused in 250 mL of water and strained.	Single dose (with 1-week washout between each arm of the study)	Both fennel and fenugreek increased feelings of fullness and decreased desire to eat food; however, there were no changes in amount of food consumed after drinking either tea compared with placebo tea.	Low
Fenugreek	[88]	Clinical trial	20 adults with hypercholesterolemia	12.5–18 g/day	4 weeks	Total cholesterol and LDL cholesterol decreased at both doses.	Low
	[89]	Clinical trial	pts with type 1 diabetes	100 g/day	10 days	Fenugreek reduced fasting blood sugar, improved glucose tolerance and reduced LDL, total cholesterol and triglycerides.	
	[90]	Clinical trial	Type 2 diabetics	15 g	Single dose	Postprandial glucose was decreased, but there was no impact on insulin or lipids.	
	[91]	Double-blind RCT	13 healthy volunteers	3 g/day	10 days	Fenugreek improved glucose tolerance and insulin sensitivity (as shown by reduction in melanin-concentrating hormone).	High
	[92]	Double-blind cross-over RCT	10 healthy volunteers and 6 pts with type 2 diabetes	Bread with 10% fenugreek	Single dose	Adding fenugreek (1 part to 9 parts of wheat flour) reduced the glycaemic response and GI of bread in both healthy volunteers and diabetics.	Low
	[93]	Clinical trial	18 pts with type 2 diabetes	10 g/day	8 weeks	FBS, TG and VLDL-C decreased (25%, 30% and 30.6%, respectively) after taking fenugreek seed soaked in hot water whereas there were no changes in lab parameters in cases who consumed it mixed with yoghurt.	Low
	[94]	Double-blind RCT	8 pts with diabetes	5.6 g in bread	Single dose	Blood glucose was not changed, but total insulin concentration decreased.	High
	[95]	Randomised cross-over study	10 healthy adults	Bread with 10% fenugreek	Single dose	Adding fenugreek reduced the glycaemic response and GI of bread.	Low
	[96]	Randomised clinical trial	12 pts with uncontrolled diabetes	2 g/day	12 weeks	Blood glucose was not changed, but fasting insulin level increased. The ratio of HDL:LDL decreased.	Low
	[97]	Parallel randomised study	48 pts with type 2 diabetes	15 g/day fenugreek powder	8 weeks	Fenugreek decreased CRP and increased superoxide dismutase. There was no effect on glutathione peroxidase activity, total antioxidant capacity, IL-6 or TNFα.	High
	[98]	Clinical trial	60 type 2 diabetics	25 g/day	24 weeks	Serum cholesterol and triglyceride were reduced.	Low
	[99]	Cross-sectional observational study	25 pts with type 2 diabetes	5 g/day	12 weeks	Fasting blood glucose was decreased by month 2. Postprandial blood glucose level was lower by month 3.	
	[100]	Parallel randomised study	50 pts with type 2 diabetes	15 g/day fenugreek powder	8 weeks	Fenugreek decreased fasting blood glucose, and liver enzymes, serum ALT and alkaline phosphatase, compared with baseline. Compared with control group, SBP, AST and irisin (a marker of metabolic health) were decreased.	High
	[101]	RCT	114 pts with type 2 diabetes	50 g/day	4 weeks	Fenugreek improved lipid metabolism.	Low
	[102]	Double-blind RCT	56 adults with borderline hyperlipidemia	8 g/day	8 weeks	TG, LDL, total cholesterol and FBG were decreased by fenugreek.	High
	[103]	Triple-blind RCT	88 pts with type 2 diabetes	10 g/day	8 weeks	Fenugreek seeds decreased FBG and HbA1c, serum levels of insulin, HOMA-IR, total cholesterol and TG and increased serum levels of adiponectin.	High
	[104]	Double-blind RCT	125 pts with type 2 diabetes	10 g/day	8 weeks	Fenugreek alone and fenugreek combined with nutrition training decreased FBG, HbA1c, BMI and waist circumference compared with placebo.	High
	[105]	Double-blind RCT	62 pts with type 2 diabetes	10 g/day fenugreek powder	8 weeks	Fenugreek improved mean FBG, HgA1C, BMI, waist circumference, DBP and quality of life	High
	[106]	Randomised cross-over study	8 healthy individuals	25 g	single dose	Fenugreek seeds reduced the rise in blood glucose and insulin caused by a meal.	Low
Garlic	[107]	Clinical trial	40 pts with MetS	100 mg/kg bodyweight crushed garlic	4 weeks	Raw crushed garlic reduced waist circumference, SBP and DBP, TG, FBG and significantly increased serum HDL cholesterol. There was no significant difference found in BMI.	Low
	[108]	Clinical trial	4 healthy adults	40 g fresh garlic	1 week	Garlic reduced the serum cholesterol and triglycerides when consumed with a high-fat diet.	Low
	[109]	Clinical trial	20 healthy individuals	3 g/day	90 days	Garlic reduced total cholesterol and LDL, but had no impact on total bacterial faecal count.	
	[110]	Single-blind randomised cross-over study	18 healthy volunteers	4.2 g	1 week	Baseline values of platelet function were within normal range in all volunteers. Platelet function was not impaired by single and repeated oral consumption of Greek tsatsiki containing raw garlic.	Low
	[111]	Randomised clinical trial	112 hyperlipidemic pts	20 g/day	8 weeks	Garlic and a combination of garlic and lemon reduced blood lipids (total cholesterol, TG and LDL) and BP, while increasing HDL.	Low
	[112]	Double-blind RCT	90 overweight smokers	2.1 g/day	12 weeks	Garlic had no effect on inflammatory biomarkers, endothelial function or lipid profile in normolipidemic subjects with risk factors for CVD.	High
	[113]	Double-blind RCT	90 pts with NAFLD	1.6 g/day	12 weeks	Garlic decreased hepatic steatosis, liver enzymes and blood lipids (total cholesterol, TG, HDL and LDL).	High
	[114]	Double-blind RCT	90 pts with NAFLD	1.6 g/day	12 weeks	Waist circumference, body fat, FBG, insulin and insulin resistance improved. Skeletal muscle mass increased and antioxidant capacity increased.	High
	[115]	Double-blind RCT	90 pts with MetS	1.6 g/day	12 weeks	Garlic increased HDL. There were decreases in waist circumference, BP, TG, insulin and appetite.	High
Garlic and coriander seed	[116]	Single-blind RCT	80 pts with hyperlipidemia	2 g/day	40 days	Garlic and coriander improved BMI, total cholesterol, HDL and LDL. Garlic powder was more effective than coriander	Low
Ginger	[117]	Double-blind RCT	160 obese children with NAFLD	1 g/day	12 weeks	Serum FBG and CRP, BMI, waist circumference, AST, hepatic steatosis, total cholesterol and LDL decreased with ginger.	High
	[118]	Placebo-controlled study	23 healthy male volunteers	1 g	Single dose	Ginger had no effect on thermoregulatory function, but increased fat utilisation in the morning.	High
	[119]	Randomised cross-over study	18 healthy volunteers	15 g raw ginger or 40 g cooked ginger	2 weeks	Ginger did not affect thromboxane production.	High
	[120]	Randomised cross-over study	10 healthy men	2 g	2 days	Ginger enhanced thermogenesis and reduced hunger and food intake.	Low
	[121]	Double-blind RCT	20 60-year-old pts with diabetes	3 g	12 weeks	Improvements found in blood glucose, insulin resistance, inflammatory and oxidative markers (CRP and MDA).	High
	[122]	Double-blind RCT	45 diabetic pts	2 g/day	10 weeks	Ginger supplementation decreased ADMA serum levels (although this change was not significantly different to placebo), but had no effect on sICAM-1.	High
	[123]	Double-blind RCT	50 pts with NAFLD	1.5 g/day of ginger	12 weeks	No difference between ginger and placebo for anthropometric measurements or liver markers. However, FBG and insulin resistance were improved by ginger. Serum lipids (total cholesterol and LDL) and CRP decreased in the ginger group.	High
	[124]	Double-blind RCT	45 diabetic pts	2 g/day	10 weeks	No effect of ginger on anthropometric measurements or NFκB	High
	[125]	Double-blind RCT	64 pts with type 2 diabetes	2 g/day	8 weeks	TNFα and hs-CRP were reduced by ginger. IL-6 was reduced by ginger compared with baseline, but not compared with placebo.	High
	[126]	Double-blind RCT	64 pts with diabetes	2 g/day	8 weeks	Ginger supplementation significantly lowered the levels of insulin, LDL-cholesterol, TG and the HOMA index and increased the QUICKI index, but had no effect on FBG, total cholesterol, HDL-C and HbA1c.	High
	[127]	Double-blind RCT	88 diabetics	3 g/day	8 weeks	Ginger improved fasting blood glucose, insulin levels and insulin sensitivity.	High
	[128]	Double-blind RCT	80 healthy obese women	2 g/day	12 weeks	Ginger decreased anthropometric measurements and appetite.	High
	[129]	Double-blind RCT	70 women with gestational diabetes	1.5 g/day	6 weeks	Ginger treatment reduced the levels of FBS, serum insulin and HOMA index, but did not affect postprandial blood sugar.	High
	[130]	Double-blind RCT	70 Obese women	2 g/day	12 weeks	Ginger reduced blood glucose, total cholesterol, TG and LDL/HDL, while increasing MDA and HDL. Body weight, waist circumference and BMI were also reduced without any difference in energy and macronutrient intake between groups.	High
	[131]	Pilot double-blind RCT	60 hypothyroid pts with normal serum TSH	1 g/day	30 days	Ginger reduced the thyroid symptom score. Additionally, weight gain, cold intolerance, constipation, dry skin, appetite, memory loss, concentration disturbance and feeling giddy or dizzy domains also improved. Ginger supplementation also led to a decrease in body weight, BMI, waist circumference, serum TSH, FBG, TG, and total cholesterol levels compared to the placebo.	High
	[132]	Double-blind RCT	70 pts with type 2 diabetes	1.6 g	12 weeks	Ginger reduced multiple markers of metabolic health (blood glucose, insulin, insulin resistance, cholesterol) and inflammation (CRP and prostaglandin).	High
	[133]	Double-blind RCT	80 Obese women	2 g	12 weeks	There was a reduction in BMI, serum insulin and HOMA-IR.	High
	[134]	Double-blind RCT	103 pts with diabetes	1.2 g/day	12 weeks	Ginger reduced fasting blood glucose and total cholesterol.	High
	[135]	Double-blind RCT	41 pts with type 2 diabetes	2 g/day	12 weeks	Ginger supplementation reduced the levels of fasting blood sugar, HbA1c, apolipoprotein B, apolipoprotein B/apolipoprotein A-I and MDA in the ginger group in comparison to baseline, as well as the control group, while it increased the level of apolipoprotein A-I.	High
	[136]	Double-blind RCT	85 pts with hyperlipidemia	3 g/day	45 days	TG and cholesterol levels were lower in the ginger group than placebo. Changes in LDL and HDL were not significant between the two groups.	High
Ginger and fenugreek	[137]	Placebo-controlled study	30 pts with coronary artery disease and 30 healthy individuals	4 g/day or 10 g one-off dose of ginger. 5 g/day fenugreek	12 weeks	Ginger did not affect platelet aggregation when given at 4 g/day, but 10 g single dose reduced platelet aggregation. Ginger had no effect on blood sugar or blood lipids. Fenugreek had no effect on cholesterol, TG or blood sugar in healthy individuals, but reduced cholesterol and triglycerides in pts with coronary artery disease and diabetes.	Low
Mustard, black pepper, ginger, horseradish	[138]	Single-blind cross-over trial	22 young, healthy males	20 g ginger or 1.3 g of black pepper	Single dose	Ginger and black pepper had no effect on appetite, energy intake or diet-induced thermogenesis.	High
Nigella seeds	[139]	Double-blind cross-over RCT	51 pts with MetS	3 g/day	8 weeks	Nigella had no effect on BP, weight, waist circumference, FBG and BMI.	High
	[140]	Double-blind cross-over RCT	51 pts with MetS	3 g/day	8 weeks	No effect of nigella on blood lipids, apolipoproteins and inflammatory factor.	High
	[141]	Double-blind RCT	30 healthy male volunteers	1 g/day	4 weeks	Nigella seeds had no effect on glycaemia or insulin. Total cholesterol and LDL-cholesterol were decreased with no effect on triglycerides or HDL-cholesterol.	Low
	[142]	Clinical trial	94 pts with type 2 diabetes	1, 2 or 3 g/day	12 weeks	Reductions in FBG, postprandial glucose, HbA1c and insulin resistance from 2 g/day. No effect on serum C-peptide or body weight.	Low
	[143]	Placebo-controlled study	35 menopausal women with MetS	1 g/day	8 weeks	No change in body weight. Nigella reduced FBG. Total cholesterol, TG and LDL were reduced. HDL change was not significant.	Low
	[144]	Randomised controlled trial	37 menopausal women with moderate risk of hyperlipidemia	1 g/day	8 weeks	Decrease in hyperlipidemia from nigella seeds.	Low
	[145]	Single-blind, non-randomised trial	114 pts with type 2 diabetes	2 g/day	one year	Nigella seed group had a significant decline in TC, LDL-cholesterol, total cholesterol/HDL-C and LDL/HDL ratios, as well as DBP, mean arterial pressure and heart rate.	High
	[146]	Double-blind RCT	73 adults with hyperlipidemia	2 g/day	6 weeks	No effects were seen for blood sugar or lipids.	High
	[147]	Double-blind RCT	40 pts with Hashimoto’s thyroiditis	2 g/day	8 weeks	*Nigella sativa* supplementation significantly reduced anthropometric variables including weight, BMI and waist circumference. Serum TSH and anti-TPO concentrations reduced while serum T3 increased in *Nigella sativa* treated group. VEGF also decreased in the nigella group.	High
	[148]	Double-blind RCT	50 pts with NAFLD	2 g/day	12 weeks	Levels of CRP and NFκB were decreased by nigella. No change in hepatic steatosis or TNFα.	High
	[149]	Double-blind RCT	50 pts with NAFLD	2 g/day	12 weeks	Reduction in glucose, insulin and insulin resistance, but no change to lipid profile. Percentage of hepatic steatosis also decreased.	High
	[150]	Double-blind RCT	40 pts with Hashimoto’s thyroiditis	2 g/day	8 weeks	Nigella seeds decreased TG, LDL, weight and BMI, and increased SOD and total antioxidant capacity.	High
	[151]	Double-blind RCT	114 pts with type 2 diabetes	2 g/day	one year	No change to BMI. Nigella led to a decrease in FBG, HbA1c and insulin resistance.	High
	[152]	Double-blind RCT	140 menopausal women with MetS	500 mg/day	8 weeks	LDL-cholesterol, TG, total cholesterol and FBG decreased.	High
	[153]	Double-blind RCT	39 men with obesity	3 g/day	12 weeks	There was no change in serum-free testosterone, FBG, TG and inflammatory markers. However, body weight, waist circumference and BP did improve.	High
	[154]	Randomised, placebo-controlled trial	74 individuals with hypercholesterolemia	2 g/day	4 weeks	Nigella seed lowered triglycerides, LDL and cholesterol compared with baseline, but had no effect on blood glucose or HDL.	High
Rosemary	[155]	Double-blind RCT	110 pts with NAFLD	4 g/day	8 weeks	There were no effects on liver enzymes, anthropometric measurements, FBG, insulin, insulin resistance and blood lipids from rosemary when compared with placebo.	High
Sage	[156]	Non-randomised cross-over trial	6 healthy female volunteers	600 mL sage tea/day	4 weeks	No effects on blood glucose, but LDL and total cholesterol decreased, while HDL increased. Lymphocyte hsp70 expression also increased.	Low
Turmeric	[157]	Single-blinded RCT	42 women with type 2 diabetes and hyperlipidemia	2.1 g/day	8 weeks	There was an improvement in body composition, lipid profile and glycemic status in the turmeric group and the aerobic training or the aerobic training plus turmeric groups compared with control group. The combined group also had lower blood lipids, blood glucose and insulin than turmeric alone group.	High
	[158]	Randomised cross-over study	14 healthy volunteers	6 g	Single dose	Turmeric increased postprandial insulin without affecting plasma glucose.	High
	[159]	Open-label, randomised clinical trial	60 diabetic pts on metformin	2 g/day	4 weeks	Turmeric reduced fasting plasma glucose, but had no effect on post-prandial glucose. Turmeric also increased glutathione, MDA and CRP compared with baseline. LDL cholesterol was decreased compared with baseline.	High
	[160]	Double-blind RCT	46 pts with NAFLD	3 g/day	12 weeks	Turmeric consumption decreased serum levels of glucose, insulin, HOMA-IR and leptin. Changes in weight, BMI and liver enzymes were not significant	High
	[161]	Randomised, single-blinded placebo-controlled trial	42 hyperlipidemic pts with type 2 diabetes	2.1 g/day	8 weeks	Turmeric alone and turmeric plus aerobic training sig decreased waist circumference, FBG, TG and BP, while HDL cholesterol increased. MetS Z score and inflammatory markers improved in both turmeric groups.	High
	[162]	Double-blind RCT	80 type 2 diabetes mellitus pts (30–70 years old)	2.1 g/day turmeric powder	8 weeks	Turmeric was found to decrease body weight, TG and total cholesterol.	High
	[163]	Double-blind RCT	64 pts with NAFLD	2 g/day	8 weeks	Turmeric reduced liver enzymes AST, ALT and GGT compared with placebo. Triglycerides, LDL, HDL and MDA decreased from baseline, but were not different from placebo.	High
	[164]	Double-blind RCT	114 pts with type 2 diabetes	1.2 g/day	12 weeks	Turmeric reduced arterial stiffness. No change was found in markers of endothelial function.	High
Turmeric and nigella seed	[165]	Double-blind RCT	250 healthy men with MetS	1.5 g/day nigella, 2.4 g/day turmeric	8 weeks	Nigella seed led to improvement in triglycerides, total cholesterol, LDL and HDL, but no change to anthropometric measures, blood glucose, BP or inflammation. Turmeric improved cholesterol, LDL and inflammation, but not TG, HDL, anthropometric measures, BP or blood glucose.	High
Turmeric and cinnamon	[166]	Double-blind RCT	48 people >60 years with prediabetes	1 g turmeric or 2 g of cinnamon	Single dose	Co-ingestion of turmeric with white bread increases working memory independent of body fatness, glycaemia, insulin or AD biomarkers. Cinnamon had no impact on working memory. Use of turmeric or cinnamon regularly had no impact on glycaemia or insulin responses to breakfast.	High
Turmeric and red pepper spice	[167]	Double-blind cross-over RCT	98 overweight or obese women	2.8 g turmeric/day	4 weeks	Turmeric does not alter oxidative stress or inflammation in overweight/obese females with systemic inflammation or cause a significant shift in the global metabolic profile.	High

AD: Alzheimer’s disease, ADMA: asymmetric dimethylarginine, ALT: alanine transaminase, AST: aspartate aminotransferase, BMI: body mass index, BP: blood pressure, CRP: c-reactive protein, CVD: cardiovascular disease, DBP: diastolic blood pressure, FBG: fasting blood glucose, FSH: follicle stimulating hormone, GER: gastric emptying rate, GGT: γ-glutamine transpeptidase, GI: glycaemic index, GIP: glucose-dependent insulinotropic polypeptide, GLP: glucagon-like peptide, HbA1c: glycated haemoglobin, HDL: high-density lipoprotein, HOMA-IR: homeostasis model assessment of insulin resistance, 2hpp: 2-h postprandial, hsp: heat shock protein, IL: interleukin, LDL: low-density lipoprotein, LH: leutenising hormone, MDA: malondialdehyde, MetS: metabolic syndrome, MMP: matrix metalloproteinase, NAFLD: non-alcoholic fatty liver disease, NFκB: nuclear factor κB, NO: nitric oxide, PCOS: polycystic ovarian syndrome, pt: patient, QUICKi index: quantitative insulin sensitivity check index, RCT: randomised-controlled trial, SBP: systolic blood pressure, sICAM: soluble intercellular adhesion molecule, Sirt: sirtuin, SOD: superoxide dismutase, TG: triglycerides, TNF: tumor necrosis factor, TPO: thyroid peroxidase, TSH: thyroid-stimulating hormone, VCAM: vascular cell adhesion protein, VEGF: vascular endothelial growth factor.

## Data Availability

The data presented in this study are openly available in FigShare at https://doi.org/10.6084/m9.figshare.24411967.

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
