# Peer review of "A Scoping Review of the Clinical Evidence for the Health Benefits of Culinary Doses of Herbs and Spices for the Prevention and Treatment of Metabolic Syndrome"

_nutrients, 2023, doi:10.3390/nu15234867_

Round 1
Reviewer 1 Report
Comments and Suggestions for Authors
The present review was to investigate whether there was consistency in the doses used and outcomes found, but consistency in the doses and outcomes found is difficult to maintain in actual research. The author's summary of relevant research is qualitative analysis, lacking quantitative analysis and description of key indicators, resulting in a lack of high-quality data support for the final conclusion, which is only a rough description but not meta-analysis for quantitative analysis. The author adopted a literature inclusion strategy based on meta-analysis, but the analysis was only qualitative, resulting in insufficient support for the conclusion from the results. I suggest that the author reorganize the article according to the requirements of the review.
1. Introduction: Metabolic syndrome refers to the abnormal indicators that need to be explained in the background to facilitate the correspondence between relevant indicators and diseases in the results
2. In the description of the results, the author should focus on spices with a large research population, while other studies with a relatively small number can be described together.
3. When summarizing the research results, the author should not mix the results of cohort studies and intervention studies to describe them together. This article summarizes the intervention results of spices on metabolic syndrome, and should not be included in intervention studies targeting normal individuals. I suggest that the author establish their own inclusion and exclusion criteria and strictly enforce them.
4. The author summarized the research on the effects of spices on indicators related to metabolic syndrome using a method similar to meta-analysis, but did not conduct quantitative analysis. The included research types are diverse and the population is complex, making it difficult to form a unified conclusion. It is recommended that the author modify it to a regular review.
5. The author found that several spices have certain health effects but lacked discussion of potential biological mechanisms.
Author Response
Thank you for your comprehensive and detailed response. A scoping review includes a thorough and systematic approach to searching with qualitatiave analysis of data. The purpose is not to demonstrate efficacy, but to show the level of evidence that exists (eg, Scoping Reviews - Systematic Reviews - LibGuides at Weill Cornell Medical College). We have included a sentence in the introduction to explain further: “The many different types of evidence available for the health benefits of herbs/spices mean that systematic review methods are not yet appropriate. This methodology uses a systematic approach to searching, but produces qualitative results to highlight available research and provide a base for determining what further research is needed.”
- Introduction: Metabolic syndrome refers to the abnormal indicators that need to be explained in the background to facilitate the correspondence between relevant indicators and diseases in the results
We have added text to the introduction to include the key metabolic biomarkers linked to metabolic syndrome and how they can be monitored to measure risk of metabolic syndrome developing or show improvements in those with metabolic syndrome.
- In the description of the results, the author should focus on spices with a large research population, while other studies with a relatively small number can be described together.
We have re-arranged the results section to have separate sections for the herbs/spices with the most studies and collate the herbs/spices with fewer studies together.
- When summarizing the research results, the author should not mix the results of cohort studies and intervention studies to describe them together. This article summarizes the intervention results of spices on metabolic syndrome, and should not be included in intervention studies targeting normal individuals. I suggest that the author establish their own inclusion and exclusion criteria and strictly enforce them.
The cohort studies were included following Google Scholar searches, rather than from the more detailed PubMed and Scopus searches. We have removed the cohort studies.
However, as we were interested in the potential for herbs/spices to prevent metabolic disease as well as treating it, it is important to include studies that investigated their impact on metabolic markers in healthy individuals, as well as the effects on individuals with disease. We have made that clearer in the introduction and methods.
- The author summarized the research on the effects of spices on indicators related to metabolic syndrome using a method similar to meta-analysis, but did not conduct quantitative analysis. The included research types are diverse and the population is complex, making it difficult to form a unified conclusion. It is recommended that the author modify it to a regular review.
We used a PRISMA extension for scoping reviews methodology and have made that clearer in the introduction and methods section
- The author found that several spices have certain health effects but lacked discussion of potential biological mechanisms.
There are biological mechanisms included in the discussion for many of the herbs/spices, as well as a summary at the end of the discussion.
For example, “Cardamom reduced swelling and downregulated inflammatory cytokines such as cyclo-oxygenase-2 (COX-2) in an animal model… A terpenoid compound from cardamom, 1,8-cineole, prevented lipid oxidation in vitro and lowered serum lipid levels in zebrafish”
“This impact on blood sugar is thought to be due to an insulin-mimetic effect and via inhibition of digestive enzymes such as α-amylase in the gastrointestinal tract [174, 176]. Cinnamon activates both PPARα and PPARγ, which would explain an effect on glycaemia”
“Larger doses of fenugreek tended to be used, as the beneficial effect was usually considered to be from the soluble fibre in the seeds”
“Garlic and its phytochemicals have been found to have anti-hyperlipidaemic activity via 3-hydroxy-3-methylglutaryl CoA (HMG CoA) inhibition and reduction of cholesterol synthesis; hypotensive activity via angiotensin-converting enzyme (ACE) inhibition, downregulation of angiotensin II and stimulation of nitric oxide; and, anti-inflammatory/anti-atherosclerotic effects via COX inhibition, decreased synthesis of thromboxane B2, decreased production of leukotriene C4 and reduction of LDL oxidation”
“The health benefits of herbs/spices are likely due to their phytochemical content and complex interactions between these molecules, other dietary components, the microbiome and the gut wall. Phytochemicals, and polyphenols specifically, have been found to impact carbohydrate absorption and metabolism, gut bacteria populations and uptake of glucose into muscle and adipose tissue”
Reviewer 2 Report
Comments and Suggestions for Authors
The manuscript “Scoping review of clinical evidence for the health benefits of culinary doses of herbs and spices for prevention and treatment of metabolic syndrome” by Mackonochie and colleagues provides an interesting outline of the available clinical and epidemiological evidence, investigating the effects of dietary achievable amounts of herbs and spices on biomarkers related to the metabolic syndrome.
The criteria and decision-making process used for reviewing the evidence are thoroughly explained, and the summary table provides a clear and easy-to-read overview of the findings.
I have a few minor comments and suggestions:
- Since the term “Metabolic syndrome” is repeated many times over the manuscript, maybe an abbreviation (such as MetS) could be used for ease of reading
- Line 101. If you have filtered the PubMed search for “Clinical Trials” as article type, how could you find the epidemiological cohort trials?
- The use of the Jadad scale is not very appropriate for trials investigating food because it penalizes the use of fresh food. Indeed, two out of five points depend on blinding, which is impossible when using fresh food (how do you blind fresh garlic or fresh ginger!?). I would at least mention this important limitation when the scoring system is explained.
- In Figure 2, I find the double charts for each biomarker to be a little redundant, and I think it would be more clear and visually effective for the reader to just keep the four charts including all studies (a, c, e and g), and possibly all four in the same page. But this is just my personal suggestion, for the authors to consider.
- Line 8: If you want to name in the abstract the databases you have used, you should include Google Scholar too.
- Line 21. There’s no full stop at the end of the line.
- Line 105. Use capital S in “Scholar”
- Table 1, there are no horizontal lines separating the sections for different spices, making it quite difficult to understand where the set of studies for the next spice begins.
- Table 1, either use “patients” or “pts”
- Table 1, either use “sig” or “significant”. I would suggest writing the word in full, or just removing it (if a result is reported, it is obvious that it was statistically significant).
- Table 1, reference 68, replace “blinde” with “blind”
- Table 1, reference 168, replace “2.1g” with “2.1g/day”
- Lines 368-380. This should be moved as caption for Figure 2. Alternatively, figure 2 caption should explain the meaning of the different letters. Or, consider replacing the letters “a” through “g” with self-explaining titles.
- Line 582. Replace “those” with “patients”
- Line 582. Replace “seen” with “tested” (or “directly investigated”, or “specifically investigated”).
- Line 588. Replace “where both fresh and dried can be used” with “which can be used both fresh and dried”.
- Line 613, delete “here”
- Line 623, replace “the remaining diet” with “the rest of the diet”
- Line 630, replace the two commas with “;”
- Line 634, I would replace “other important lifestyle changes” with “other important dietary and lifestyle changes”.
Author Response
Thank you for your constructive feedback and thorough correction of the manuscript.
- Since the term “Metabolic syndrome” is repeated many times over the manuscript, maybe an abbreviation (such as MetS) could be used for ease of reading
Good suggestion, thanks. We have made that change
- Line 101. If you have filtered the PubMed search for “Clinical Trials” as article type, how could you find the epidemiological cohort trials?
These were added from Google Scholar searching, but we have decided to remove all the cohort studies for consistency with the search terms. The discrepancy of including cohort studies was also pointed out by the other peer reviewer.
- The use of the Jadad scale is not very appropriate for trials investigating food because it penalizes the use of fresh food. Indeed, two out of five points depend on blinding, which is impossible when using fresh food (how do you blind fresh garlic or fresh ginger!?). I would at least mention this important limitation when the scoring system is explained.
This is an important point and we have added it in the methods section when discussing the Jadad scale.
- In Figure 2, I find the double charts for each biomarker to be a little redundant, and I think it would be more clear and visually effective for the reader to just keep the four charts including all studies (a, c, e and g), and possibly all four in the same page. But this is just my personal suggestion, for the authors to consider.
This is a good point, however, we have decided to keep all the charts in to allow comparison with high-quality and all studies.
- Line 8: If you want to name in the abstract the databases you have used, you should include Google Scholar too.
Done
- Line 21. There’s no full stop at the end of the line.
Thanks, corrected.
- Line 105. Use capital S in “Scholar”
Corrected
- Table 1, there are no horizontal lines separating the sections for different spices, making it quite difficult to understand where the set of studies for the next spice begins.
Corrected. We've included grid lines for the whole table and it is much easier to read. Thanks
- Table 1, either use “patients” or “pts”
Corrected. We have used pts and included an abbreviation in the legend.
- Table 1, either use “sig” or “significant”. I would suggest writing the word in full, or just removing it (if a result is reported, it is obvious that it was statistically significant).
We’ve removed all sigs and only included essential significants.
- Table 1, reference 68, replace “blinde” with “blind”
corrected
- Table 1, reference 168, replace “2.1g” with “2.1g/day”
Corrected
- Lines 368-380. This should be moved as caption for Figure 2. Alternatively, figure 2 caption should explain the meaning of the different letters. Or, consider replacing the letters “a” through “g” with self-explaining titles.
This has been moved to the caption.
- Line 582. Replace “those” with “patients”
corrected
- Line 582. Replace “seen” with “tested” (or “directly investigated”, or “specifically investigated”).
Corrected
- Line 588. Replace “where both fresh and dried can be used” with “which can be used both fresh and dried”.
corrected
- Line 613, delete “here”
Corrected
- Line 623, replace “the remaining diet” with “the rest of the diet”
Corrected
- Line 630, replace the two commas with “;”
We couldn’t see two commas
- Line 634, I would replace “other important lifestyle changes” with “other important dietary and lifestyle changes”.
Corrected
Round 2
Reviewer 1 Report
Comments and Suggestions for Authors
The author has actively responded to relevant comments and clearly described the limitations of the article. This review is a summary of existing research on spice health, and although there are some shortcomings, it is still very valuable.
Comments on the Quality of English LanguageThe author has actively responded to relevant comments and clearly described the limitations of the article. This review is a summary of existing research on spice health, and although there are some shortcomings, it is still very valuable.
Reviewer 2 Report
Comments and Suggestions for Authors
The authors have satisfactorily addressed all my concerns about the manuscript. I recommend publication in the present form.